# Optimizing the Retrieval of the Vital Status of Cancer Patients for Health Data Warehouses by Using Open Government Data in France

**DOI:** 10.3390/ijerph19074272

**Published:** 2022-04-02

**Authors:** Olivier Lauzanne, Jean-Sébastien Frenel, Mustapha Baziz, Mario Campone, Judith Raimbourg, François Bocquet

**Affiliations:** 1Analytics Department & Data Factory, Institut de Cancérologie de l’Ouest, F-44805 Nantes-Angers, France; olauzanne@gmail.com (O.L.); mustaphabaz@hotmail.com (M.B.); 2Oncology Department, Institut de Cancérologie de l’Ouest, F-44805 Nantes-Angers, France; jean-sebastien.frenel@ico.unicancer.fr (J.-S.F.); mario.campone@ico.unicancer.fr (M.C.); judith.raimbourg@ico.unicancer.fr (J.R.); 3Center for Research in Cancerology and Immunology Nantes-Angers, INSERM UMR 1232, Nantes University and Angers University, F-44307 Nantes-Angers, France; 4The Law and Social Change (DCS) Laboratory, UMR CNRS 6297, F-40000 Nantes, France

**Keywords:** vital status, record linkage, homonyms, deterministic linkage, fuzzy matching, health data warehouse

## Abstract

Electronic Medical Records (EMR) and Electronic Health Records (EHR) are often missing critical information about the death of a patient, although it is an essential metric for medical research in oncology to assess survival outcomes, particularly for evaluating the efficacy of new therapeutic approaches. We used open government data in France from 1970 to September 2021 to identify deceased patients and match them with patient data collected from the Institut de Cancérologie de l’Ouest (ICO) data warehouse (Integrated Center of Oncology—the third largest cancer center in France) between January 2015 and November 2021. To meet our objective, we evaluated algorithms to perform a deterministic record linkage: an exact matching algorithm and a fuzzy matching algorithm. Because we lacked reference data, we needed to assess the algorithms by estimating the number of homonyms that could lead to false links, using the same open dataset of deceased persons in France. The exact matching algorithm allowed us to double the number of dates of death in the ICO data warehouse, and the fuzzy matching algorithm tripled it. Studying homonyms assured us that there was a low risk of misidentification, with precision values of 99.96% for the exact matching and 99.68% for the fuzzy matching. However, estimating the number of false negatives proved more difficult than anticipated. Nevertheless, using open government data can be a highly interesting way to improve the completeness of the date of death variable for oncology patients in data warehouses

## 1. Introduction

Comprehensive clinical data warehouses are crucial tools for improving overall understanding of cancer diseases and patient care in oncology [1]. Indeed, with an efficient and up-to-date data warehouse, it is possible to benefit from many advantages, such as medical analysis, anomaly detection, and classification and prediction of diseases [2]. Electronic Medical Records (EMR) and Electronic Health Records (EHR) are increasingly used for real-world studies in oncology, which require accurate dates of death to assess survival outcomes [3]. When patients die outside of health care facilities, hospitals often do not register the patients’ death. The absence of such essential information has a negative impact on the medical research in oncology [4].

In general, the evaluation of the efficacy of a drug treatment aims to measure the magnitude of the effect of the drug relative to the clinically relevant comparator, most often in terms of morbidity–mortality, quality of life, and safety. As in other fields, in oncology, the most robust endpoint, and considered as the gold standard, is overall survival (OS). The OS is the length of time from either the date of diagnosis or the start of treatment for a disease, such as cancer, that patients diagnosed with the disease are still alive. The OS is universally recognized as being unambiguous, unbiased, with a defined endpoint of paramount clinical relevance, and positive results provide confirmatory evidence that a given treatment extends the life of a patient. Even if clinical trialists relentlessly attempt to devise more easily measured, cost-effective, and readily available event-driven endpoints as predictive surrogates of a definitive outcome, such as OS—and thus reduce the time with which clinical trials deliver definitive results—OS remains the reference endpoint in oncology [5,6,7]. Real-world data sources containing dates of death can be used to calculate long-term OS, but only if vital status is reliably captured. The value of these external real-life data sources is invaluable and can address a critical need in cancer research to identify truly effective treatments, procedures, and therapeutic approaches for which there was only partial evidence of effectiveness at the time of their evaluation in clinical trials. This is the purpose of this study, which was conducted at the Institut de Cancérologie de l’Ouest (ICO), integrated cancer center in western France.

Some studies were carried out to retrieve patient’s vital status for hospitals [8,9,10]. Fournell et al. [10] did this in France with death registry data, and they used probabilistic linkages based on a methodology developed by Jaro [11]. They also used phonetic codes adapted to the French language, and their results were evaluated on reference data produced from the “*Registre National d’Identification des Personnes Physiques*” (RNIPP French Identification Register of Individuals). Some other studies focus on analyzing online obituaries by “scraping” (i.e., technique based on computer scripts that allows content to be extracted from one or more websites in a fully automatic way) [12,13]. In a 2018 study, Sylvestre et al. mention using the French national death registry, but they do not explain exactly how it is combined with online obituaries [12].

A British study conducted in 2019 by Doidge et al. mention date of death retrieval as a possible objective for record linkage [14]. Record linkage refers to linking entities between two datasets; this can be divided into three categories: the deterministic linkage, probabilistic linkage, and alternative linkage methods [15]. Deterministic linkage uses predefined rules to link entities; it is most often based on simply exact matches between fields, but it can be arbitrarily complex [16]. Probabilistic linkage usually refers to the methods developed by Newcombe et al. [17] and formalized by Fellegi and Sunter [18]. It associates each pattern of matching fields to the likelihood that two records are a match and combines these likelihoods into a score that can either be the real likelihood that two records are a match, or an indicator correlated with this probability when the fields are not statistically independent. Alternative linkage methods exist, and they usually use machine learning [19] or Bayesian models [20,21,22]. Wilson [19] shows that probabilistic record linkage is equivalent to a naive Bayes classifier and that it can be replaced by other classifiers; he also shows that even a single layer neural networks can, in some cases improve record linkage compared to standard probabilistic linkage.

In order to complete the dates of death in our data warehouse, we needed to select an external updated database with the most personal information that was the same as in our internal database, such as first name, surname, sex, date, and place of birth, plus the information we wanted to extract, such as vital status and date of death [23]. In France, The National Institute for Statistics and Economic Studies has been in charge of producing, analyzing, and publishing official statistics since 1946 [24], and its data has a very high percentage of reliability [25]. It continuously collects information on all individuals who die each week, through the intermediary of the town halls. The latter transmit the death certificates to the INSEE, either in paper form or digitally. The INSEE then uses this information to produce the CSV (comma-separated values) files that we used as our data source. These files also include French citizens deceased abroad [26].

The aim of this study was to refine a methodology for linking the national death data registry to hospital patients data, and to evaluate the quality of the data collected. We decided to use deterministic linkage because of its simplicity, and we tried to develop methodologies to evaluate our results without reference data. We applied two different matching algorithms to reliably retrieve the dates of death. We started by using an exact matching algorithm, which recovers the date of death if an individual is found in the national registry with the same name, given name, place of birth, and date of birth as in the patient data. The second algorithm was a fuzzy matching algorithm, which can find an individual in the national registry despite small spelling differences between the national registry and the hospital data warehouse. We tried to estimate the probability of mistaking a patient for a homonym, which we used to estimate the number of false links. We then investigated the false negatives to understand why they were missed by our matching algorithms [27].

## 2. Materials and Methods

### 2.1. Data Sources and Architecture

We used two data sources: the ICO patient data warehouse and the INSEE data of deceased persons [26]. Patient data contains information on 166,156 patients dating from 2015 to November 2021, with 12,667 patients that had a date of death. We can assume that those dates of death are mostly correct ones, but we know that some dates of death are missing. The INSEE data contains 26,095,782 rows from 1970 to September 2021, which was all the available data in the national registry at the time we performed this study. The information in common between the two databases was last name, first given name, sex, date of birth, and place of birth. We used both an exact matching algorithm and a fuzzy matching algorithm to find patients’ dates of death when they were present in the national registry. We always tried to evaluate the results and especially the dates of death present in the hospital data warehouse and missing from the linked data to refine or increase the accuracy of our matching algorithms. As shown in Figure 1, the data processing begins with pre-processing the hospital patients data and death register data to ensure that fields are similarly formatted, text fields are stripped of leading and trailing spaces, names are converted to upper case and hyphens are removed, the sex field is converted to “M” or “F”, and the dates are converted to 8 digits numbers. After the pre-processing, we applied the matching algorithms to link the patients’ data with the death register data. We used this matched dataset to estimate the number of false negatives. In this context, false negatives are patients who are identified as still alive but are deceased. To estimate the number of false positives, we calculated the risk of patients having homonyms. False positives are patients who are identified are deceased but are still alive.

### 2.2. Organisation of the Study

A senior and a junior data scientist were involved in data matching. Three oncologists and the head of the ICO’s data factory and analytics department were involved in validating all the steps of the methodology and interpretation of the results. In terms of inter-reliability, by design, the records matched by the exact matching algorithm are always matched by the fuzzy matching algorithm, and we verified that it is the case in practice. The study was conducted over 3 months, between November 2021 and January 2022.

### 2.3. Matching Algorithms

#### 2.3.1. Exact Matching Algorithm

We used the fields of surname, first name, sex, date of birth, and place of birth from each database to develop an exact matching algorithm. This involved a pre-processing task on these fields in order for each field to have the same format. A first data matching was carried out with the condition of having the same name, first given name, date of birth, and place of birth. Following this linkage, a first evaluation was carried out in order to examine the results obtained. This evaluation pointed out that certain dates of death already present in the patient database had not been retrieved from the national death register database or were different from the dates present in the national registry. This was caused by two main factors: the first was that some compound names were marked with a hyphen on one of the databases, e.g., “Jean-Francois”, and with a space on the other, e.g., “Jean Francois”. The second reason is that some birthplaces were incorrectly entered in one of the databases. In order to get around these two problems, the following procedure was followed: the hyphens were replaced with spaces to avoid inconsistencies, and we checked that the given name field in the national register started with the given name string in the hospital data warehouse, because the hospital database only contains the first given name. We also improved this linkage by excluding dates of death that dated from before the last visit of a patient.

#### 2.3.2. Fuzzy Matching Algorithm

To increase the number of matched deceased patients, we decided to develop a fuzzy matching algorithm. The objective was to accept a certain amount of spelling errors and still accept the match. We calculated an error score and two rows were considered a potential match if the error score was lower than 1. To compare two character strings, we used the Levenshtein distance divided by the length of the longest character string to get a score between 0 and 1. The Levenshtein distance, sometimes also called the edit distance, is the number of single character edits (insertions, deletions, substitutions) that are needed to go from one character string to another; it is frequently used to identify similar character strings [28,29].

This string distance score indicates how similar two character strings are; the score is 0 if the character strings are exactly the same and 1 if they could not be more different from each other. This score was then multiplied by a weight that was different for each field and which represented how important the field was for identification. The values of those weights were chosen so that identified miss links of the exact matching algorithm could match. The weight of the name is 5 and the weight of the given name is 3. The choice of those weights was also motivated by understanding which potential matches should be excluded by those weights. For example, a weight of 5 for the name field means that if you have two records whose fields are identical except for names that differ by one letter from each other, then the records will match if the names have 5 letters or more. An example of error score values is given in Table 1.

For the INSEE code representing the place of birth, the Levenshtein distance was not used because many matching records had the correct first two digits (department number) and completely different values for the remaining digits. Instead, when the codes differed, the error score was increased by 0.5 if the first two digits were identical and by 1 if they were different. If the sex was different, the error score was increase by 0.5, so that sex alone can never be the only cause for not matching two records. Similarly, to the exact matching algorithm, dates of death that happened before the last visit of the patient were excluded; then, if there were several potential matches for the same patient, the one with the lowest error score was chosen. The years of birth collected in the national registry data containing only a year, or a year and a month, were deleted for the exact matching algorithm. To obtain more matches, we set the dates to the first day of the month, or the first day of January if only the year was present. We chose these dates because they often matched what was entered in the hospital data warehouse. To increase the speed of the algorithm, we decided not to try all pairs two by two, but to match pairs by searching in two different types of block. The first type of blocks used the date of birth combined with the first four letters of the name as a key, and the second type used the date of birth combined with the first letter of the name and the first three letters of the given name.

### 2.4. Evaluation

#### 2.4.1. False Negatives

A patient could be a false negative if the linkage process missed a link between the patient and the death register or if the death register was missing a death record concerning the patient. Therefore, we intended to evaluate false negatives by looking at the number of dead patients in the hospital data warehouse that were not matched with a deceased person in the national death registry. This included both false negatives because of missing links and false negatives because of missing records. Because we lacked a reference dataset, we decided to assume that the ratio of dead patients that were not matched in the national death registry was the same for dead patients whose date of death was in the hospital data warehouse and for those whose date of death was missing.

#### 2.4.2. Homonymity Rates

In this study, exact homonyms refer to people who can be misidentified by the exact matching algorithm, which includes people who not only share the same name and given name but also the same date of birth, place of birth, and sex. Similarly, fuzzy homonyms are people that can be mistaken for each other by the fuzzy matching algorithm. The homonymity rate for a certain year was the number of homonyms born in a certain year compared to the total population born that year. Meanwhile, the population born that year was the set of people that should find themselves in the national death register, which contains people who die in France as well as French people who die abroad. Estimating the homonymity rate required a significant amount of data cleaning because the national death register contained many duplicates, with some of the duplicates containing typing errors. We used the date of death as an additional variable to distinguish between homonyms and duplicate entries. Once the homonyms born in a certain year were found in the death register, the issue remained that some people born that year were still alive. We needed to consider that the people already in the national register are a sample of all the people born that year who will eventually end up in the database.

If we have a population of size, P, with a homonymity rate of r and we take a representative sample of the population of size S and assume that homonyms are only in pairs, then a homonym present in the sample will only have an S/P chance of also having its pair in the sample and of being perceived as a homonym when only looking at the sample. Therefore, the apparent rate of homonymity, $r^{\prime }$, will be lower in the sample than the rate in the total population because $r^{\prime }=r\times S/P$, which allows us to deduce the real rate, r, from $r^{\prime }$:$$r=r^{\prime }\times P/S$$

If the assumption about homonyms only going in pairs is wrong, this formula will still be on the safe side as it will overestimate the real homonymity rate. The total population, P, was estimated by first looking at national birth statistics [30], which gives the number of people born in a given year in metropolitan France, $P^{\prime }$, and then looking for the number of people born that year already in the death register, D, and the number of people born that year in the death register that were not born in metropolitan France, $D^{\prime }$; we used the following formula:$$P=P^{\prime }\times \left(1+D^{\prime }/D\right)$$

We used this methodology to calculate the homonymity rate for each birth year between 1900 and 1970.

#### 2.4.3. False Positives

As the homonymity rate is the chance that one person can be mistaken for another by the matching algorithm, if a person has a homonym, there is a priori a one in two chance that a link concerning this person is a false link. We thus used half of the homonymity rate as the linkage precision. The real linkage precision would be higher because, if the homonym died before the patient first visited the hospital, the match could be excluded. Moreover, the mortality rate of a cancer patient is probably higher than the mortality rate in the general population. There are two reasons why a patient could be a false positive: a false link and a false date of death. We know that the quality of the data in the national death registry is not perfect. Therefore, even if the linkage is correct, there still could be an incorrect data in the national death registry. Even though we were not able to evaluate the quality of the data quantitatively in this study, we did check the cases where the date of death in the hospital data warehouse was significantly different from the date of death in the national data (difference of more than 62 days, which is the maximum number of days contained in two months). However, we were only able to identify the correct date of death in some of the cases.

## 3. Results

### 3.1. Homonyms

The homonyms were tested year by year for years of birth between 1900 and 1970. We limited our study of homonyms to the patients born during the period 1900–1970 because the death register contains too few people born after 1970 and people before 1900 have little relevance to our study” Figure 2 shows that the homonymity rate for the exact matching has an upward trend and there was a significant amount of noise between the years 1955 and 1970. This is simply because fewer people born between those dates are deceased, leading to a smaller sample size for finding homonyms.

Figure 3 shows that the homonymity rate for the fuzzy matching has no real trend.

The homonymity rate for the patients was calculated as a weighted average of the homonymity rate for each patient born between the years 1900 and 1970. This gave a homonymity rate of 0.106% for the exact matching algorithm and 0.656% for the fuzzy matching algorithm.

### 3.2. Matching Results

Of the 166,156 patients in the hospital data warehouse, only 12,667 originally had a date of death. As shown in Table 2, the exact matching algorithm identified 26,193 matches, which more than doubled the number of dates of death for the hospital patients. The fuzzy matching algorithm identified 37,434 matches, which more than tripled the number of patients with a date of death in the hospital data warehouse.

The estimated linkage precision is 99.95% for the exact matching algorithm and 99.67% for the fuzzy matching algorithm. As expected, the fuzzy matching algorithm is less precise than the exact matching algorithm, but it has fewer missed links.

### 3.3. False Negatives

Trying to calculate the number of false negatives by using the assumption that the patients in the hospital data warehouse with a date of death were representative of the dead patients gave the following results: a recall of 89.10% for the exact matching algorithm and 97.01% for the fuzzy matching algorithm. These results were not compatible with the rest of the data: the exact matching algorithm found 30.02% less results than the fuzzy matching algorithm, and the fuzzy matching algorithm had less than 1% false positives. Taking false positives into account, and assuming that the fuzzy matching algorithm had no false negatives, the recall of the exact matching algorithm should be less than 70.16%.

We thus formulated an ad hoc hypothesis to estimate the false negatives. Our best guess is that the number of patients whose death was present in the hospital data warehouse and not found by a matching algorithm was correlated in a linear manner to the number of dead patients the matching algorithm would miss. This hypothesis gives recall values of 65.73% for the exact matching and 94.58% for the fuzzy matching.

### 3.4. Patients with Different Dates of Death

For 18 patients, the date of death in the national database was significantly different from the date of death in the hospital data warehouse, with a difference of more than 62 days, which is the maximum number of days contained in two months. We manually looked at those patients, and for 11 of them we were able to find some information to indicate which date of death was correct. In all but one case the correct date of death was the one in the national registry. The study of the homonymity rates predicted approximately 13.86 false positives for the patients present in the hospital data warehouse with the exact matching algorithm and about 122.83 false positives with the fuzzy matching algorithm.

## 4. Discussion

Prior to this work, of the 166,156 patients included in the ICO data warehouse, we knew the dates of death for only 12,667 of them. The exact matching algorithm identified 26,193 matches, which allowed us to more than double the number of dates of death for our patients. The fuzzy matching algorithm identified 37,434 matches, which more than tripled the number of patients with a date of death in our data warehouse. While there is still room for improvement, these results show the importance of matching data from data warehouses with external data sources to improve the completeness of dates of death in our warehouse. It should be remembered that a large proportion of patients die outside of the hospital and that this explains why information on their dates of death is not automatically collected in hospital data warehouses.

### 4.1. False Negatives

One of the most surprising results of this study was how hard it was to estimate the number of false negatives without reference data. We assumed that the patients with a date of death present in the hospital data warehouse would at least be somewhat representative of all the deceased patients. However, the number of false negatives predicted by this assumption for the exact matching algorithm was three times lower than expected. One possible explanation is that patients whose date of death was present in the hospital data warehouse were often patients who died in hospital. In this case, the hospital was at least in part responsible for producing the death certificates, which could be either produced directly from information contained in the hospital data warehouse or produced more accurately than for patients who did not die in the hospital. Not being able to estimate the number of false negatives was a real issue. Not being able to estimate the number of false negatives was a real issue, and using an ad hoc hypothesis only provides uncertain results. To obtain a better estimate for the number of missed matches we would need to create a “gold standard” subset manually and evaluate our linkage on this subset [15,31,32,33]. It would also be interesting to check the completeness of the death registry, possibly by comparing it with mortality rates in relation to age and birth rates.

### 4.2. Patients with Different Dates of Death

It may seem a little surprising that both the exact matching and fuzzy matching had the same number of patients with different dates of death. One explanation is that the exact matching algorithm matched most of the patients whose date of death was present in the hospital data warehouse, while the fuzzy matching algorithm always produced the same link as the exact matching algorithm when the latter produced a link. Another possible explanation is that the number of false positives was overestimated because the assumption that there was a one in two chance of mistaking a patient for a homonym when the patient had a homonym was overly conservative.

### 4.3. Estimated Precision

To estimate the number of false positives, we assumed that there was a one in two chance of mistaking a patient for his or her homonym. This really overestimated the risk because there was a chance that the homonym died when we knew the patient was still alive, making us exclude the match. Furthermore, the risk of a cancer patient dying was probably higher than the risk in the general population. Further work could be done in that direction to improve the precision of the estimates.

### 4.4. Homonyms

One possible improvement for the homonymity rate would be to compute it separately for men and women. This might be more precise because it would take into account different trends in given names. It would also improve the correctness of the assumption that the people born in a certain year present in the national death registry were a representative subset of the people born that year. At present, because of their longer life expectancy, women may be slightly underrepresented in the national registry. In this study, we never looked at individual characteristics as a way of better estimating the risk of having a homonym. Studying the popularity of given names could factor into the homonymity risk. For example, for the year of birth 1900, 44 exact homonyms were found, and of those 44 homonyms, 42 of them had the given name “Marie”.

### 4.5. Probabilistic Linkage

While some studies argue that both deterministic linkage and probabilistic linkage have their own strengths [34,35,36], Doidge et al. advocate the idea that probabilistic linkage is superior in almost all cases [16]. We used deterministic linkage because of its greater simplicity. It has indeed produced fewer false links and more missed links than probabilistic linkage in a similar case [10]. However, it is hard to say if this is because Fournel et al. [10] used phonetic codes or simply because they chose a different trade-off between false links and missed links [16]. We do not think this difference is caused by probabilistic linkage having intrinsically more false links. The fuzzy matching equation is very similar to an equation that could have been produced by a probabilistic approach, especially with a Fellegi–Sunter linkage extended with approximate field comparators [37,38,39]. A possible explanation could be a sort of anchoring effect because deterministic linkage usually begins (and often ends) with an exact matching algorithm, which has very few false links.

### 4.6. Strengths and Limitations of the Study

Not having reference data to estimate the quality of the linkage record was the most problematic issue that we faced. It forced us to use the death registry data to estimate the number of homonyms and the rate of false positives. Estimating the number of false negatives turned out to be more complicated than anticipated; we ended up having to use an ad hoc hypothesis to have a result at all. It would be interesting to create a reference dataset to see how correct this ad hoc hypothesis was.

Compared to the work of Fournel et al. [10], which uses probabilistic record linkage, our approach is easier to set up, especially when only using exact matching. Moreover, we developed ways to estimate the accuracy of our matching algorithm without using reference data.

Our fuzzy matching algorithm used both very generic and very specific elements, depending on the fields. The use of a normalized Levenshtein distance in the error score calculation is applicable to any record linkage project based on text fields, on the other hand the error score associated with the place of birth is very specific to this project.

The information that a match is not only a fuzzy match but also an exact match is kept in the warehouse database, so that if we want to minimize the number of false positives, we can use exact matches if necessary. To improve on this idea, it would have been possible to make the fuzzy matching more or less restrictive by keeping the error score in the database and applying a filter on this error score when necessary.

## 5. Conclusions

By using two matching algorithms on the national death registry, we were able to extensively enrich the ICO data warehouse. The hospital database originally contained 12,667 dates of death, the exact matching algorithm found 26,193 dates of death and 14,907 new dates of deaths, more than doubling the number of dates of death present in the hospital data warehouse. The fuzzy matching algorithm was developed to be able to match incorrectly spelled data and was able to find 37,434 dates of death, more than tripling the number of dates originally present in the hospital data warehouse. Studying the risk of patients having homonyms allowed us to estimate the number of false positives, assuring us that the risk of misidentification was low, both for the exact matching algorithm and the fuzzy matching algorithm, with estimated precisions of 99.96% and 99.68%, respectively. Moreover, considering the patients for whom the national death registry and the hospital data warehouse were contradictory, in most cases it was the hospital data warehouse that was at fault. Finally, we found it surprisingly difficult to estimate the number of deceased patients that the matching algorithms missed. We noticed that the patients whose dates of death were already present in the hospital data warehouse were not representative of the dead patients whose dates of death were missing, so that only an ad hoc hypothesis allows us to estimate the number dates of death that are missing. The date-of-death data obtained was integrated into the ICO data warehouse to optimize the conduct of real-life studies in cancer. Overall, open government data can be used to improve the completeness of the date of death variable for hospital data warehouses.

## Figures and Tables

**Figure 1 ijerph-19-04272-f001:**
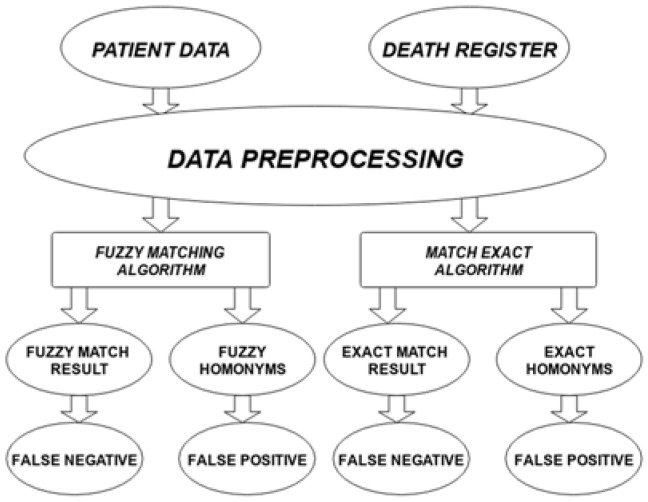
Data Preprocessing: Fuzzy matching algorithm and Match exact algorithm. Patient data and national registry data went through pre-processing, then the matching algorithms performed the linkage between the two pre-processed databases. The matching algorithms were also used to study homonyms.

**Figure 2 ijerph-19-04272-f002:**
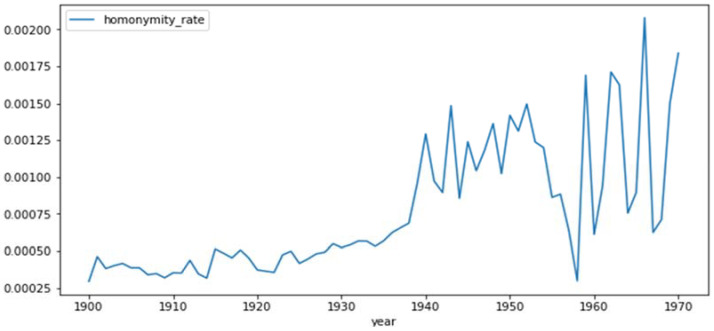
Homonymity rates for the exact matching between 1900 and 1970. The exact homonymity rate (risk that an individual is indistinguishable from another for the exact matching algorithm) produced from the national death registry for each year of birth between 1900 and 1970.

**Figure 3 ijerph-19-04272-f003:**
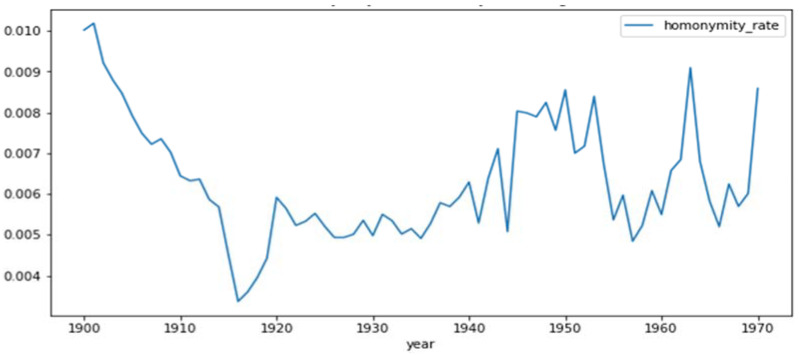
Homonymity rates for the exact matching between 1900 and 1970. The fuzzy homonymity rate (risk that an individual is indistinguishable from another for the fuzzy matching algorithm) produced from the national death registry for each year of birth between 1900 and 1970.

**Table 1 ijerph-19-04272-t001:** Possibility of a match by applying the tolerance system using an error score based on the Levenshtein distance *.

Patient Data		Death Register		Match	Score
First Name	Last Name	First Name	Last Name		
Maxim	Dupond	Maxime	Dupond	YES	0.5
Maxime	Dupont	Maxime	Dupond	YES	0.833
Maxim	Dupont	Maxime	Dupond	NO	1.333

* The Leveshtein distance is the number of single character edits (insertions, deletions, substitutions) that are needed to go from one character string to another.

**Table 2 ijerph-19-04272-t002:** Results obtained after the application of both the exact matching algorithm and the fuzzy matching algorithm.

	Exact Matching	Fuzzy Matching
Total number of matches	26,193	37,434
Matches not in hospital data warehouse	14,907	25,146
Matches already present in hospital data warehouse	11,286	12,288
Matches with a difference of more than 62 days	18	18
Matches missing compared to the hospital data warehouse	1381	379
Estimated linkage precision (%)	99.95	99.67
Estimated number of false links	13.86	122.83

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
