# Peer review of "Optimizing the Retrieval of the Vital Status of Cancer Patients for Health Data Warehouses by Using Open Government Data in France"

_ijerph, 2022, doi:10.3390/ijerph19074272_

Round 1

Reviewer 1 Report

The article shows how the use of open government data can be a way to improve the completeness of clinical data. Specifically, the paper describes an approach that uses two different algorithms to associate the date of death with deceased cancer patients. This study aims to develop a methodology to link the national registry of death data to hospital patient data and to assess the quality of the data collected. Specifically, French open government data were used to identify deceased patients and compared with patient data collected from the Institut de Cancérologie de l'Ouest data warehouse. The topic of the paper is relevant, some related work is considered. The paper is well-written, organized, and clear.

It might be helpful to include a short introduction in paragraph 2 which would describe the architecture and functionality generally. For example, the preprocessing functionality could be briefly described, or what is meant by false positive or false negative for the different results obtained by the two algorithms shown in figure 1. In this way, it is possible from the first reading to have a global understanding of the proposal and then analyze it further in the following paragraphs.

In the paragraph "Matching algorithms" the rules used in data preprocessing are briefly described. It might be useful to show, later in the Discussion section or in the conclusions, which of these rules can be considered general (or generalizable and in what way) and which can only be used in this specific case of matching between registers (or language-related).

It would be interesting to describe the process to find the best weights in calculating the Levenshtein distance for the proposal in the "Matching algorithms" section.

For the result section, it would be useful to identify among the links created the set of associations that represent the intersection between the two algorithms. For example, to understand how many matches of the exact algorithm have not been identified by the fuzzy algorithm and vice versa. This information could lead to improving the proposal for identifying links.
In this way, as future work, a hybrid approach could be developed, aiming at the advantages of the two solutions and obtaining better results.

I found the use of open government data to enrich clinical data and thus foster research very interesting. Probably a future effort can be aimed at generalizing the approach to apply it to different datasets.

Reviewer 2 Report

The topic that author addressed is of huge importance for both academia and practicioners.  The information about the death of a patient is often not related to the EHR. The research included huge sample of related data. However, certain parts of the paper should be improved and refined.

The literature overview should be extended and consider broader literature.  What is the main contribution of the paper? How does the paper conform to the existing literature? What new does it bring?

Discussion should be improved in the great extent. What are the implications of the research? How the results could be used in practice and research?

Reviewer 3 Report

The article addresses the need of implementing electronic health and medical records about the death of oncologic patients by matching data retrieved from 2 databases. Although the original claim that providing more complete information about the death of patients respond to an urgent need of assessing the survival rate, this concept is only briefly mentioned in the introduction, and not highlighted anymore throughout the manuscript. I believe some methodological concerns need to be addressed, and the clinical relevance of study need to be stressed more. 

I suggest addressing the following corrections.

Introduction: I consider it enough explicative and easy to follow. However, I would encourage the authors to stress more upon the importance of this study. It is only mentioned that this can help in the medical analysis and to assess survival outcomes. But why do we need to know the survival rate? This is what legitimate the study, so I would particularly highlight this passage.

In line 68-69, I would use the same term as "alternative linkage methods" when you described it, so that the reader doesn't get lost. Moreover, I suggest providing a more extensive description of the third method, as done with the previous two. 
I encourage the authors to provide a reference to the statement (line 75-76) that "its data has a very high percentage of

I would indicate that the aim of the study was to apply a methodology, rather  than develop a methodology. Indeed, in the following sentence, the authors indicate that they used a deterministic linkage, so they are not developing any new method. 

Methods:

I believe some flaws need to be addressed in this section: who and how many researchers perform the matching of the data? Was the matching process verified for inter-reliability? When the research was performed? How long did it take? Is this study approved by any Institutional Review Board, is it involved dealing with sensitive data? 
Also, are those individuals that die abroad included in the analysis? I would clarify that the information of patients that die not in the country of France are transmitted to the INSEE. Specify this in the introduction (in the paragraph where you are describing the process of transmitting information to INSEE or in the methodology, as appropriate). 
- the acronym of ICO is not spelled out in the manuscript. It is only in the abstract. I would clarify what it is. Moreover, in my opinion, it would be easier to follow what the ICO is and why it was used it that information was added in the introduction. 
- I would suggest providing a reference of the use of Levenshtein distance in this field of forensic identification. (maybe this can fit your need:  doi: 10.1016/j.fsigen.2021.102594)
- line 126: remove "to"  
- why was the Levenshtein distance not used for the place of birth? Is the error score given to sex? Please support this choice with rationale or with reference, as appropriate.
- line 224: please, specify better what "few cases" means in terms of exact number. 

Results: 
It is not clear why the homonyms were tested between 1900 and 1970, as the data retrieved were between 1970 and 2021. Please address this doubt. 
-  I would move the interpretation of the results of line 274-281 to the discussion, as in this section the authors should just report the results, with no other personal interpretation

Discussion:
please, start the section of the discussion by reporting the findings of the study as first paragraph, and highlight the importance of the study (e.g., what is the clinical relevance of this study?)
- I suggest reporting the results of the ad hoc analysis in the result section, as the discussion section should not introduce any new findings, but only discuss the observation in light of what is there in the literature.
- in the first paragraph of false negative, please provide a comparison with other system in the literature to find if the % of false negative is way higher that what is already known in the literature. Also, in all the paragraphs I encourage the author to restate their findings in light of the existing literature.
- please, include a paragraph with strength and limitations of the study at the end of the discussion 
- also, where these data available for future studies on the survival rate for individuals with cancer, which was the original claim? 

Round 2

Reviewer 3 Report

I believe the authors thoroughly and extensively addressed all the comments. I believe this revision improved the soundness and the scientific robustness of the paper. Good job to the authors.